# Hardware Efficient Massive MIMO Systems with Optimal Antenna Selection

**DOI:** 10.3390/s22051743

**Published:** 2022-02-23

**Authors:** Shenko Chura Aredo, Yalemzewd Negash, Yihenew Wondie Marye, Hailu Belay Kassa, Kevin T. Kornegay, Feyisa Debo Diba

**Affiliations:** 1School of Electrical and Computer Engineering, Hawassa University, Hawaasa 05, Ethiopia; 2School of Electrical and Computer Engineering, Addis Ababa University, Addis Ababa 1176, Ethiopia; yalemzewdn@yahoo.com (Y.N.); yihenew.wondie@aait.edu.et (Y.W.M.); 3Department of Electrical & Computer Engineering, Morgan State University, Baltimore, MD 21251, USA; hailu.kassa@morgan.edu; 4Scool of Electrical Engineering and Computing, Adama Science and Technology University, Adama 1024, Ethiopia; feyisa2006@yahoo.com

**Keywords:** antenna selection, beamforming, Digital to Analogue Conversion, energy efficiency, massive MIMO, mmWave

## Abstract

An increase in the number of transmit antennas (M) poses an equivalent rise in the number of Radio Frequency (RF) chains associated with each antenna element, particularly in digital beamforming. The chain exhibits a substantial amount of power consumption accordingly. Hence, to alleviate such problems, one of the potential solutions is to reduce the number of RFs or to minimize their power consumption. In this paper, low-resolution Digital to Analogue Conversion (DAC) and transmit antenna selection at the downlink are evaluated to favour reducing the total power consumption and achieving energy efficiency in mMIMO with reasonable complexity. Antenna selection and low-resolution DAC techniques are proposed to leverage massive MIMO systems in free space and Close In (CI) path-loss models. The simulation results show that the power consumption decreases with antenna selection and low-resolution DAC. Then, the system achieves more energy efficiency than without low-resolution of DAC and full array utilization.

## 1. Introduction

Massive MIMO (mMIMO) is a large-scale MIMO device that is becoming more common in wireless communications and which scales up traditional MIMO by orders of magnitude [1]. It considers multi-user MIMO in which a base station has hundreds and thousands of antennas supporting multiple single-antenna terminals at the same time and frequency resources.

A device with a large number of antenna elements increases the connection reliability, spectral quality, and radiated energy efficiency. Each antenna element is linked to a single RF chain at the base station, which comprises mixers, analogue-to-digital converters (ADC), and amplifiers [2]. Furthermore, the increase in the number of antennas and associated RF chains at the base station will result in physical restrictions, complexity, and expense [3]. According to [4], RF chains are responsible for approximately 50–80% of a base station’s total transceiver power consumption.

The hardware complexity and power consumption of DACs increase exponentially with the number of quantization bits as the Base Station (BS) antenna elements increase. Thus, using a low-resolution DAC is a promising option [5]. The energy consumption of the power amplifier is also influenced by conversion from analogue to digital and digital to analogue (ADC/DAC), phase shifters, and power amplifiers. Though the digital beamforming system provides a high data rate, the energy consumption becomes excessive since the transceiver system uses the same number of antennas as the chains.

In contrast, a hybrid beamforming system uses fewer RF components, which can be used to offer comparable spectral efficiency to a digital beamforming system while being more energy-efficient [6]. Even though hybrid beamforming is the solution as a technique employing a small number of RF chains, cutting down some numbers among the entire array is one of the questions left as an open issue. Due to this, working on low-resolution DAC and antenna selection has been used as one of the power reduction techniques for a system with an extensive array of RF components.

Most recent literature studies have concentrated on performance analysis for large MIMO uplinks using analogue-to-digital converters with limited resolution. In [7], the effect of signal detection schemes on uplink MIMO systems’ energy efficiency with low-resolution analogue-to-digital converters were evaluated. There have been an increasing number of studies for the case of downlink transmission with low-resolution DACs. The Energy Efficiency (EE) of hybrid transmitters with DACs quantized based on additive quantization noise was explored in [8,9,10,11].

Consequently, a sub-optimal method was utilized to build an optimum hybrid precoder based on the Additive Quantization Noise Model (AQNM). It also compares quantized digital precoders to hybrid ones with a wholly or partially linked phase-shifting network of active/passive phase-shifters. The challenges of downlink precoding for multi-user MIMO on a narrow-band system with low-resolution DACs at a BS are investigated in [9].

Nonetheless, most researchers have proposed low-resolution DAC for hybrid beamforming, where limited baseband units are used and with low power demand; there should be an equivalent solution for digital beamforming. Since digital beamforming is known for its high capacity at the expense of increased power consumption, we propose antenna selection with low-resolution DAC as a viable option for addressing the inherent hardware complexities and power consumption.

Over the last few decades, various antenna selection techniques and algorithms have been investigated for the classic MIMO. In [12], basic selection algorithms for realistic detectors were used to examine error rate-based performance evaluations. The studies in [13,14,15] promoted capacity-oriented selection criteria like the greedy algorithm and convex optimization. The authors in [16] presented an antenna selection technique (AS) with a minimal level of complexity that picks antennas that minimize constructive user interference. When the transmitter uses precoders in conjunction with a matched filter, the suggested AS algorithm outperforms systems that use a more complicated channel inversion method. The work in [17,18] aimed to remove the destructive portion of the interference, which was established by the connection between the substreams of a modulated Phase Shift Keying (PSK) scheme.

According to the authors in [19], singular value decomposition was utilized to offer a new Euclidean Distance based Antenna Selection technique (EDAS) for antenna selection in spatial modulation systems that has lower computational complexity than exhaustive search. Furthermore, the Symbol Error Rate (SER) approaches a full search when the number of received antennas grows. Therefore, in comparison with the past and current research trends, the authors of [20] stated that there is still considerable interest in mmWave-based massive MIMO antenna selection with manageable complexity, more energy efficiency, and optimal data rates in recent years.

In this paper, a system with transmit antenna selection for massive MIMO-enabled BS is considered after low-resolution DAC is applied. The procedure is divided into three parts: First, the EE of an entire array device is evaluated at the cell edge using a fixed power allocation technique. In this case, the optimal number of BS antennas $(M^{⋆}$) at which the EE reaches its maximum is determined among the total number of BS antennas (*M*) using the initial access condition. Second, the minimum Signal-to-Noise Ratio (SNR) to be found at the cell edge is used as a threshold value to search the optimal number further when users move from the cell edge to outskirts or centre positions. In this scenario, $M^{⋆}$ is considered to be a maximum number of elements.

Due to the position changes, $M^{⋆}$ is transformed to ($M^{o}$), representing the number of selected antennas while the transmit power adaptively changes due to distance changes in mobility. Following the determination of $M^{o}$, the subset of antennas with the best channel conditions are chosen, and EE is assessed using spatial selectivity at sub-6 GHz and mmWave frequency ranges. Finally, EE is evaluated by integrating a selection algorithm with a low-resolution DAC.

The main contributions of this paper are stated as follows:We introduce an energy-efficient downlink antenna selection technique for mobile and static users. The proposed technique considers two-phase selection:Optimal number of BS antennas ($M^{⋆}$) at which the energy efficiency graph becomes maximum and starts declining, is determined as, $M^{⋆}=M\left(EE_{i}=EE_{max}\right)$. For this, the following assumptions are used:–A maximum number users a BS can support is assumed, and all users are to be at the cell edge distances.–All BS antennas and RF components are employed to determine total downlink power consumption according to (Equation 34).–The channel is assumed to be random, and we consider fixed SNR (${\bar{\gamma }}_{k}$), which is the average of least SNR values from several random channel generations for cell edge as in (Equation 18). A minimum SNR value is considered to accommodate the worst-case in which the channel is in deep fading.Next, user mobility-based selection is made. In this case, our selection algorithm incorporates the exhaustive searching method to select a group of elements with the best channel gain as in (Equation 21) and (Equation 23). The double section also reduces the number of search combinations and computational complexity. Again, since double selection using algorithms one and two minimizes the number of RF components directly associated with the antenna elements in the case of digital beamforming, the power consumption is substantially reduced and makes the system energy efficient.In comparison to prior methods, our proposed algorithm lowers the computational complexity of the transceiver system.We design a heuristic and simple formulation of antenna selection to evaluate the performance for mMIMO at sub-6 GHz and mmWave bands with CI and FS path-loss models.We introduce an energy-efficient and optimal DAC resolution algorithm for massive MIMO systems.Finally, by integrating our novel algorithms, the effect of selection on the EE was evaluated with low resolution and typical DAC.

The rest of the work is structured as follows: A system model for mMIMO beamforming and array geometry is defined in Section 2. After the propagation model is explained in Section 3, antenna selection and power consumption models are followed in Section 4 and Section 5, where results and analysis are presented. Finally, our conclusions are drawn in Section 6.

## 2. System Model and Description

A downlink massive MIMO system with digital beamforming is considered. As shown in Figure 1, the system consists of digital switches that are associated with each BS antenna serving simultaneous users with multiple data streams. Both DAC resolution and antenna selection are performed on the basis of digital beamforming. As shown in Figure 1, the BS has a Uniform Rectangular Array (URA) geometry with λ/2 spaced *M* antennas, where λ is the signal wavelength, and mutual coupling between antenna elements is ignored. Inside the cell, k single-antenna devices transmit data to the BS simultaneously using the same time-frequency resources.

## 3. Propagation Model and Analysis

Unlike typical low-frequency channels, mmWave channel propagation characteristics are no longer affected by Rayleigh fading, and its spatial selectivity is restricted due to significant path loss in free space [20]. As a result, the channel vector for $\hat{w}$th user and $\hat{m}$th antenna element is given by the product of Equations (Equation 2)–(Equation 5) as follows:(1)$$h_{\hat{w},\hat{m}}=\vec{G}\Omega \tilde{A}\varpi ,$$
where Equations (Equation 2)–(Equation 5) are as stated below.
(2)$$\vec{G}=\sqrt{p_{k}}{\left[\begin{array}{c} {\hat{F}}_{rx,\hat{w},\bar{\beta }}\left({\bar{\beta }}_{k},ZoA,\delta_{k},AoA\right)\\ {\hat{F}}_{rx,\hat{w},\bar{\beta }}\left({\bar{\beta }}_{k},ZoA,\delta_{k},AoA\right) \end{array}\right]}^{T}$$
(3)$$\Omega =\left[\begin{array}{cc} e^{j{\hat{\Phi }}_{k}^{\bar{\beta }\bar{\beta }}} & \sqrt{\Theta_{k}^{-1}}e^{j{\hat{\Phi }}_{k}^{\bar{\beta }\delta }}\\ \sqrt{\Theta_{k}^{-1}}e^{j{\hat{\Phi }}_{k}^{\bar{\beta }\delta }} & e^{j{\hat{\Phi }}_{k}^{\phi \delta }} \end{array}\right]$$
(4)$$\tilde{A}=\left[\begin{array}{c} {\hat{F}}_{tx,\hat{m},\bar{\beta }}\left({\bar{\beta }}_{k},ZoD,\delta_{k},AoD\right)\\ {\hat{F}}_{tx,\hat{m},\phi }\left({\bar{\beta }}_{k},ZoD,\delta_{k},AoD\right) \end{array}\right]$$
(5)$$\varpi =e^{\left(2\pi \lambda^{-1}\left({\hat{r}}_{rx,k}^{T}.{\hat{d}}_{rx,\hat{w}}\right)\right)}e^{\left(2\pi \lambda^{-1}\left({\hat{r}}_{rx,k}^{T}.{\hat{d}}_{tx,\hat{m}}\right)\right)}$$

### 3.1. Channel Model

The power associated with the *k*th user terminal is denoted by $p_{k}$, and the field patterns for $\hat{m}$th BS antenna and $\hat{w}$th user terminal in the direction of elevation angle, δ and azimuth angle, $\bar{\beta }$ are given by ${\hat{F}}_{tx,\hat{m},\delta }$, ${\hat{F}}_{tx,\hat{m},\bar{\beta }}$, ${\hat{F}}_{rx,\hat{w},\delta }$, and ${\hat{F}}_{rx,\hat{w},\bar{\beta }}$ for both elevation and azimuth directions, respectively. The arrival–departure of elevation and azimuth angles are represented as $\delta_{k},AoA$/$\delta_{k},AoD$, and ${\bar{\beta }}_{k},ZoA$/${\bar{\beta }}_{k},ZoD$ for elevation, and ${\bar{\beta }}_{k},ZoA$, ${\bar{\beta }}_{k},ZoD$, $\delta_{k},AoA$, and $\delta_{k},AoD$ for azimuth locations and $\Theta_{k}$ in (Equation 3) is the cross-polarization energy ratio.

For four distinct polarization combinations, the random starting phases are ${\hat{\Phi }}^{\bar{\beta }\bar{\beta }}$, ${\hat{\Phi }}^{\bar{\beta }\delta }$, ${\hat{\Phi }}^{\delta \bar{\beta }}$ and ${\hat{\Phi }}^{\delta \delta }$. The spherical unit vectors of the transmitter and receiver, given in Cartesian coordinates, are denoted by the symbols ${\hat{r}}_{tx,k}$ and ${\hat{r}}_{rx,k}$ where ${\hat{r}}_{i,k,\ell }={\left[sin\bar{\beta }cos\delta \;sin\bar{\beta }sin\delta \;cos\bar{\beta }\right]}^{T}$ for i∈[tx,rx]. The position vectors of the transmit and receive antenna components are ${\hat{d}}_{tx,\hat{w}}$ and ${\hat{d}}_{rx,\hat{w}}$, respectively, and therefore the uplink channel vector for a single terminal is formulated as
(6)$$h_{\hat{w}}=\left[h_{\hat{w}1}\;\;h_{\hat{w}2}\ldots h_{\hat{w}\hat{m}}\right].$$

The index *k* represents a user, and *w* represents the index of the antenna element of *k*th user. Since a user has only a single antenna in multi-user MIMO or massive MIMO systems, *w*th user antenna element can be used for the *k*th user. However, a user terminal can have more than one antenna in LTE or classical MIMO systems. Hence, the *w*th antenna of a user is the same to *k*th user.

### 3.2. Array Steering Vector

The array steering vectors for uplink and downlink are provided by $\nu_{r}\left({\bar{\beta }}_{k}^{r},\delta_{k}^{r}\right)$ and $\nu_{t}\left({\bar{\beta }}_{k}^{t},\delta_{k}^{t}\right)$, respectively. According to the URA antenna geometry of Figure 1, the azimuth-elevation plane is also given by
(7)$${\hat{A}}_{URA}\left(\bar{\beta },\delta \right)={\hat{\nu }}_{az}\left(\delta ,\bar{\beta }\right){\hat{\nu }}_{el}{\left(\delta ,\bar{\beta }\right)}^{T},$$
where $\nu_{az}\left(\delta ,\bar{\beta }\right)$ and $\nu_{el}\left(\delta ,\bar{\beta }\right)$ are steering vectors. The maximized uplink received signal due to beamforming weight and the general forms of both planes are stated in (Equation 8) and (Equation 9).
(8)$$\varphi_{k}=\nu_{k}\left(\bar{\beta },\delta \right)x=\sum_{k=1}^{K}\varphi_{k}{\hat{s}}_{k}$$

The general form of both azimuth and elevation planes can further be expressed as [21]
(9)$${\hat{\nu }}_{i}\left(\delta ,\bar{\beta }\right)=\frac{1}{\sqrt{M_{i}}}{\left[1,e^{j\varrho_{i}},\ldots ,e^{j\left(M_{i}-1\right)\varrho_{i}}\right]}^{T},i\in \left[az,el\right].$$
where $M_{i}$ represents the number of BS antennas in the azimuth and elevation planes, $\varrho_{az}=\xi dsin\left(\bar{\beta }\right)cos\left(\delta \right)$, $\varrho_{el}=\xi dsin\left(\bar{\beta }\right)sin\left(\delta \right)$, ξ=2π/λ, *d* is the distance between the elements and λ is wave length of the signal for *M* number of antennas. The array form of URA steering vector’s component is $\frac{1}{M_{az}M_{el}}$, where $M_{az}$ and $M_{el}$ are the azimuth and elevation directions of the superimposed signal components of the antenna array, respectively.
(10)$$\nu \left(\delta ,\bar{\beta }\right)=f_{1}{\left[1,\ldots ,e^{2j\varrho_{az}},\ldots ,e^{j\left(M_{az}-1\right)\varrho_{az}+\left(M_{el}-1\right)\varrho_{el}}\right]}^{T},$$
where $f_{1}=\frac{1}{M_{az}M_{el}}$. In digital beamforming, a single radio frequency chain is required for each antenna element, resulting in high power consumption and complex architecture, where an RF chain includes a down-converter, low-noise amplifier, digital to analogue converter, analogue to digital converter, and others. Therefore, for many BS antennas, the power consumption of mixed-signal components, such as high-resolution DACs and ADCs is higher, which makes the assignment of a separate RF chain for each antenna highly inefficient.

Though hybrid beamforming is one way to attain this goal, by deploying beamforming in both the digital and analogue schemes, its capacity is less than digital. Since beamforming is accomplished at the baseband frequency in digital [22], this paper aims to reduce power overheads in digital beamforming due to several RF chain components, which account for a large amount of power consumption in cellular communication.

Figure 1 presents a digital beamforming transmitter with RF URA configurations. As shown in the figure, equal number of RF chains and number of antennas are present in digital beamforming. In every symbol duration, the vector form of the data symbol can be stated as $\ddot{d}=\left[{\ddot{d}}_{1},{\ddot{d}}_{2},\ldots ,{\ddot{d}}_{k}\right]$. Then, multiplying *k*th user symbol by its beamforming vector $\;b_{k}\;\in \;{∁}^{N\times 1}$ to form a transmit vector that corresponds to the *k*th user as $x_{k}=b_{k}{\ddot{d}}_{k}$, where *N* is the number of RF chain components. Finally, the added transmitted matrix, $\bar{\ddot{d}}=\sum_{i=1}^{k}x_{i}$, pass through the RF chain components consisting of a DAC for converting the digital samples to analogue signals and transmitting them to all antennas via power amplifiers.

At the transmitter, several DACs are utilized to convert the transmission to the analogue domain. For the quantization noise of the DACs, a linear model approximation is adopted as [6]
(11)$$Q\left(u_{i}\right)=\sqrt{1-\rho b_{i}}+\epsilon_{i},$$
where *Q*(.) is a uniform scalar quantization for the *i*th RF chain and $\rho b_{i}=\frac{\pi \sqrt{3}}{2}2^{-2b_{i}}$ is the quantization distortion parameter for bit resolution equal to $b_{i}$. The input *u* is assumed to be Gaussian distributed, $\epsilon_{i}$ is the quantization noise independent of *u*, and $\epsilon ∼CN(0,\sigma^{2})$, where $\sigma^{2}$ is the noise variance. Extending the argument to the massive MIMO scenario, this is approximated as
(12)$$Q\left(u\right)\;\approx \;cu+\epsilon ,$$
where *c* is a diagonal matrix with values based on each RF chain’s DAC resolution, and $\epsilon \in CN(0,\sigma^{2}I)$, where *I* is N×N identity matrix. Approximating the linear quantizer, we may find $x_{k}\in C^{MN\times 1}$, and, as a result, the broadcast signal can be stated as follows:(13)$$x=H_{N}Q\left(u\right)=H_{N}\left(cu\;+\;\epsilon \right)=H_{N}\;cu+H_{N}\;\epsilon ,$$
where *M* and *N* are the total numbers of antennas and RF chain components, respectively, and $H_{N}$ is the channel matrix due to the number of antennas equal to the number of RF chains, *N*. The total power consumption of DAC as a function of bits is expressed as
(14)$$P_{dac}\left(b\right)=c_{1}fb+c_{2}\left(2^{b}\right),$$
where $c_{1}$ and $c_{2}$ are static and dynamic power consumption of DAC, and *f* and *b* are operating frequency and random number of bits, respectively.

### 3.3. Signal Model

Before user equipment establishes a physical connection with a BS, an initial access procedure uses reference signals. These reference signals are used for channel estimation and equalization. For our antenna selection process, we use Sounding Reference Signal (SRS) to be transmitted from a mobile terminal(uplink). SRS is used to provide the channel characteristics of a user to a BS so that the BS uses this information to allocate resources for user terminals.

In (Equation 15), the uplink reference signal that is received by a BS is represented as
(15)$$Y_{j}=\sum_{i=1}^{k_{s}}h_{ij}x_{i}w_{i}+z_{j},$$
where *j* represents any antenna element on the BS, $k_{s}$ is the number of signal sources, $h_{ij}$ is the channel between *k* and *j*, element $z_{j}$ is additive noise at the BS, and $w_{i}$ is beamforming weight. After channel estimation is done using SRS of the uplink, a BS starts sending data to user equipments through physical downlink shared channel. Therefore, the downlink signal to be received at the uplink is given by
(16)$$y_{k}=\sqrt{\frac{p_{t}M}{K}}H_{k}W_{k}x_{k}+\sqrt{\frac{p_{t}M}{K}}\sum_{i=1,j\neq k}^{K}H_{k}W_{i}x_{i}+n_{k},$$
where the first term in the right side is the desired signal, the second term is multi user interferences, and the third is user’s additive noise. $H_{k}$, $W_{k}$ and $x_{k}$ are the channel matrix, beamformer weight and downlink signal corresponding to k user, respectively.

Therefore, disregarding the interference part in Equation (Equation 16) due to assumption of a single BS designed with precoding or digital beamforming, the capacity at the cell edge that is to be maintained as a threshold is given by
(17)$$\varsigma_{th}=B\sum_{k=1}^{K}log_{2}\left(1+\frac{p_{r}{\left|H_{k}W_{k}\right|}^{2}}{N_{o}B}\right).$$
where $p_{r}$ is the received signal power, and, considering large and small scale fading, it can be evaluated as
(18)$$\varsigma_{th}=B\sum_{k=1}^{K}log_{2}\left(1+\frac{p_{t}{\left|H_{k}W_{k}\right|}^{2}}{\Gamma (D_{max})N_{o}B}\right)=B\sum_{k=1}^{K}log_{2}\left(1+{\bar{\gamma }}_{k}\right)$$
where ${\bar{\gamma }}_{k}$ is the minimum SNR as a function of the total transmit power of a BS and path loss at the cell edge $(\Gamma \left(D_{max}\right)$ (which is calculated as in (Equation 19)), $N_{o}$ is the noise spectral density, and *B* is the channel band width. This threshold capacity also depends on $|H_{k}W_{k}{|}^{2}$, which changes with the number of users and BS antennas.

### 3.4. Mobile Location and Positioning

Identifying a mobile position in today’s cellular networks is a critical problem. The angle of Arrival (AoA), Time of Arrival (ToA), and Global Positioning System (GPS) are among the techniques used. In general, there are three methods for determining the location of a mobile terminal: satellite positioning, cellular network-based positioning, and indoor positioning. The trilateration method is used to calculate a mobile’s location using a relative position of a BS. Unlike the triangulation process, which requires the angle of each user for position tracking, only the distance between the BS and each user is required in this case [23].

### 3.5. Close-In (CI) Path Loss Model

The CI model is based on Friis and Bullington’s fundamental radio propagation concepts, wherein the values are 2 for free space and 4 for the asymptotic two-ray ground bouncing model. This provides insight into path loss as a function of the environment since base station towers are tall and inter-site distances for specific frequency bands are several kilometres. Previous Ultra High Frequency (UHF)/microwave models employed a close-in standard distance of 1 km or 100 m [24].

The CI 1 m reference distance, as proposed in [25], is a suitable recommended norm that relates the real transmit power or PL to a usable distance of 1 m. Standardization to a 1 m reference distance simplifies dimension and model comparisons, provides a consistent description of the Path Loss Exponent (PLE), and allows for quick and straightforward route loss estimates without the need for a calculator [26].

Using power control mechanisms, user terminals nearer to the BS are allocated lower power than those on the outskirts to control interference and fairness. The CI path-loss model is a generic frequency model that explains large-scale path loss at all applicable frequencies in a specific context. The dynamic range of signals perceived by users in a commercial system becomes significantly lower than the equation for the CI model, which is formulated as [24]
(19)$$PL_{CI}{\left(.\right)}_{dB}=PL_{FS}\left(f,1m\right)+10nlog_{10}\left(d\right)+\chi_{\hat{\sigma }}^{CI},$$
where $n=\frac{\sum (\bar{D}A)}{\sum ({\bar{D}}^{2})}$, $A=PL_{CI}{\left(.\right)}_{dB}-PL_{FS}\left(f,1m\right)$, $\bar{D}=10log_{10}d$ denotes a single model criterion, the PLE, *d* is the transmitter-receiver separation distance *d* starting from 1 m, and (.) represents frequency and distance parameters. The free space path loss, $PL_{FS}{\left(f,1\right)}_{dB}$ at a 1 m distance from a station and carrier frequency *f* is given as
(20)$$PL_{FS}\left(f,1m\right)\left[dB\right]=20log10\left(4\pi /\lambda \right),$$
where λ is the wavelength of the signal. The CI model includes an intrinsic frequency interdependence of path loss in the 1 m $PL_{FS}$ value, and it only has one parameter compared to the Alpha-Beta-Gamma (ABG) or (α, β, and γ) model, where α and γ are coefficients showing the dependence of path loss on distance and frequency, respectively, and β is an optimized offset value for path loss in dB. ${\hat{\sigma }}^{CI}=\sqrt{\sum \chi_{\hat{\sigma }}^{CI^{2}}/T}$, where *T* is the number of data points and $\chi_{\hat{\sigma }}^{CI}$ is large-scale signal fluctuations due to the CI pathloss model.

## 4. Antenna Selection and Power Model

### 4.1. Antenna Selection

The number of antennas to be chosen is determined by adjusting an appropriate amount of transmit power to be radiated during the selection process. The number of antenna elements to be selected accounts for the system complexity [27,28,29].

Trilateration is used to pinpoint a user’s position so that the main beam can focus only on the target region, thus, reducing leakage. The transmit power adapts the user’s location changes as a function of distance due to user mobility. In this case, instead of using all the arrays and wasting resources, a certain number of transmit antennas can be decreased adaptively as user nodes become closer to the centre of the BS. For this, we consider the minimum SNR at the cell edge as a threshold value. In contrast to when allocating maximum transmit power based on edge distance, the BS only allocates power proportional to the reduced distance, resulting in only a few antennas being activated. Therefore, the first optimal number of antennas for cell-edge users $\left(M^{⋆}\right)$ is selected as follows
(21)$$M^{⋆}=\max_{\iota \in M,\;\varsigma \in [\varsigma_{th},\varsigma_{max}]}\left(EE\left(M\left(\iota \right)\right)\right)$$
where
(22)$$EE\left(M\left(\iota \right)\right)=\frac{log_{2}(real(det\left(I+\left(\frac{{\bar{\gamma }}_{k}}{M(\iota )}\right)\right)))}{{℘}_{tot}\left(\iota \right)},$$
and ${℘}_{tot}\left(\iota \right)$ is found using (Equation 34) and ι∈[1,M], where ${℘}_{tot}\left(\iota \right)$ and M(ι) are the total power and total number of antennas as a function of each iteration. Next, adaptive selection follows as the users move from cell edge to cell centre or the outskirts and given by
(23)$$M^{o}=\frac{(\sum (\Gamma (r))/k)M^{⋆}}{\Gamma (D_{max})}.$$

Substituting for $\Gamma (D_{max})$ from (Equation 18),
(24)$$M^{o}=\frac{(\sum (\Gamma (r))/k)M^{⋆}{\bar{\gamma }}_{k}N_{o}B}{p_{t}{\left|H_{k}W_{k}\right|}^{2}}.$$

Finally, using factorial permutations as (M⋆Mo)=M⋆!Mo!(M⋆−Mo)!, the antennas with the best channel gains are chosen from the list, and this minimizes complexity as compared to $\frac{M!}{M^{o}!\left(M-M^{o}\right)!}$.

The terms Γ(r) and $\Gamma (D_{max})$ in (Equation 23) refer to the CI path loss of users at any arbitrary distance and maximum distance, respectively. The number of RF chains *N* becomes the same as the number of antennas to be selected $M^{o}$ in the case of changes in channel condition or user position, in this case, in particular for digital beamforming. The selection process for the whole system is stated in algorithms one and two below.

In addition to selecting branches with the best channel gain and pilot test at the initial access, this section also aims to demonstrate the computational complexity at the receiving end in floating operation points (flops). The complexity order O, which is the number of times the capacity evaluation with the best channel gains, is formulated as O((MMselected)) and its complexity level is shown in Table 1.

The table states the combinational permutation of the algorithms, which we compare with that of [21], which accounts for n^(MMo), where $\hat{n}=M^{2}+2M^{o2}+M^{o}$ and $\ell^{⋆}$ and ⊔ are the deducted elements due to selection and additional iterations due to incorporation of low resolution DAC, respectively. According to [28,29], the computational complexity due to the selection process is shown in (Equation 27) and (Equation 28), respectively.
(25)$$O_{1}\left(.\right)=16n^{3}+{\hat{n}}^{2}\left(24M^{2}+40M+24-24M^{o2}-24M^{o}\right),$$
(26)$$O_{2}\left(.\right)=\hat{e}+20\left(M^{2}+M-{M^{o}}^{2}-M^{o}\right),$$
(27)$$O_{3}\left(.\right)=O_{1}\left(.\right)+O_{2}\left(.\right),$$
(28)$$O_{4}\left(.\right)=\hat{n}({M^{o}}^{2}\left(M^{3}\left(M+1\right)\right),$$
where $\hat{e}=\hat{n}\left(34M^{2}+44M-36{M^{o}}^{2}-34M^{o}\right)$ and (.) denotes $\left(M,M^{o}\right)$.

The above algorithm selects the optimal number of antennas considering cell edge users at initial access utilizing the reference signal. The reference signal is only used for connection setup and determining the optimal number of antennas for static users at the cell edge. Therefore, the optimal number of antennas $M^{*}$ to be used is found using at a point in which the EE graph starts declining. All the components use only a limited amount of power, which is assumed to be 15% of its connection power.

### 4.2. Capacity and Power Consumption Model

According to [19], the down-link sum capacity is given by
(29)$$\varsigma_{full}=log_{2}\left[det\left(I+\frac{\bar{\gamma }}{M}HH^{‡}\right)\right],$$
where $\bar{\gamma }$ is the average SNR per receiver, *I* is the k×k identity matrix, superscript ‡ denotes the Hermitian transpose, and *H* is the channel matrix of entire antenna array elements. The antennas that maximize the capacity are now selected so that
(30)$$\varsigma_{sel}=\underset{s(\bar{H})}{max}\;log_{2}\left[det\left(I_{k}+\frac{\bar{\gamma }}{M^{o}}\bar{H}{\bar{H}}^{‡}\right)\right],$$
where $\bar{H}$ is created by deleting $M-M^{o}$ rows from *H*, and $s(\bar{H})$ denotes the set of all possible $\bar{H}$ (whose cardinality is (MMo)), and *K* is the total number of user terminals.
(31)$$\varsigma_{DPC_{\iota }}=\max_{P_{\iota }}\;\log_{2}det\left(I+\bar{\gamma }k\left({\bar{H^{‡}}}^{\left(M^{o}\right)}\right)P_{\iota }{\bar{H}}^{\left(M^{o}\right)}\right),$$
where Pι denotes a fraction of the transmit power for a single antenna element, and *K* is the total number of user terminals. Throughput will be further enhanced with Dirty Paper Coding (DPC) by selecting antennas with better gains after iteration over all possible combinations. In the process of selecting those columns from a full array of $H_{\iota }$, an M×M matrix ▽ with a binary diagonal element is introduced.
(32)$${▽}_{i}=\left\{\begin{array}{cc} 1, & \;\mathrm{Select}\\ 0, & \mathrm{otherwise}, \end{array}\right.$$
showing whether the *i*th element is chosen and achieving $\sum_{i=1}^{L}=M^{o}$. According to Sylvester’s identity, det (I+UJ)=det(I+JU), and the DPC total rate can be rewritten as in (Equation 31) as
(33)$$\varsigma_{DPC_{\iota }}=\max_{P_{\iota }}\;\log_{2}det\left(1+\bar{\gamma }kP_{\iota }{\bar{H}}_{\iota }^{}▽{\left({\bar{H}}_{\iota }^{}\right)}^{‡}\right),$$
where $\sum_{i=1}^{k}P_{\iota ,i}=1$. The desired ▽ is discovered by increasing the average DPC sum rate.

We can categorize the total power before and after low resolution DAC and selection as
(34)$${℘}_{tot}=P_{LO}+P_{PA}+M\left(2P_{DAC}+2P_{M}+2P_{LF}+P_{HB}\right)$$
(35)$${\vec{℘}}_{tot}=P_{LO}+P_{PA}+M\left(2{\vec{P}}_{DAC}+2P_{M}+2P_{LF}+P_{HB}\right)$$
(36)$${℘}_{tot}^{o}=P_{LO}+P_{PA}+M^{o}\left(2P_{DAC}+2P_{M}+2P_{LF}+P_{HB}\right)$$
(37)$${\vec{{℘}^{o}}}_{tot}=P_{LO}+P_{PA}+M^{o}\left(2{\vec{P}}_{DAC}+2P_{M}+2P_{LF}+P_{HB}\right)$$
where ${℘}_{tot}$, ${\vec{℘}}_{tot}$, ${℘}_{tot}^{o}$, and ${\vec{{℘}^{o}}}_{tot}$ are the total power before both low resolution and selection, after low resolution and no selection, with selection and no resolution, and with both selection and low resolution, respectively. Again, $P_{LO}$ denotes the power consumption of the local oscillator, and $P_{PA}$ is the power consumption of the power amplifier. $P_{M}$, $P_{LF}$, and $P_{HB}$ are the mixer power, low pass filter power, and hybrid with buffer power, respectively.
(38)$$EE_{FA}={\left(\frac{\varsigma }{{℘}_{tot}}\right)}_{M}\approx \frac{log_{2}\left[det\left(I+\frac{\bar{\gamma }}{M}HH^{‡}\right)\right]}{{℘}_{tot}}.$$
(39)$$EE_{Se}={\left(\frac{\varsigma }{{℘}_{tot}}\right)}_{M_{selected}}\approx \frac{log_{2}\left[det\left(I+\frac{\bar{\gamma }}{M^{o/*}}HH^{‡}\right)\right]}{{℘}_{tot}^{selected}}.$$
(40)$$EE_{alg_{1+2}}\approx \frac{log_{2}\left[det\left(I+\frac{\bar{\gamma }}{\left(\frac{M\sum_{i=1}^{k}\alpha_{t}/k}{p_{t}}\right)}HH^{‡}\right)\right]}{{℘}_{tot}^{o}}.$$
(41)$$EE_{alg_{1+2+3}}\approx \frac{log_{2}\left[det\left(I+\frac{\bar{\gamma }}{\left(\frac{M\sum_{i=1}^{k}\alpha_{t}/k}{p_{t}}\right)}HH^{‡}\right)\right]}{{\vec{{℘}^{o}}}_{tot}}.$$
(42)$$=\frac{log_{2}\left[det\left(I+\frac{\bar{\gamma }}{\left(\frac{M\sum_{i=1}^{K}\alpha_{t}/k}{p_{t}}\right)}HH^{‡}\right)\right]}{P_{LO}+P_{PA}+M^{o}\left(2{\vec{P}}_{DAC}+2P_{M}+2P_{LF}+P_{HB}\right)}.$$
(43)$$EE_{DPC}=\frac{\varsigma_{DPC_{l}}}{{\vec{{℘}^{o}}}_{tot}}$$
where $\varsigma_{DPC_{l}}=\max_{P_{l}}\;\log_{2}det\left(I+\bar{\gamma }k\left({\bar{H^{‡}}}^{\left(M^{o}\right)}\right)P_{l}{\bar{H}}^{\left(M^{o}\right)}\right)$. In the *EE* equations listed above, (Equation 38) and (Equation 39) belong to energy evaluations for full array antennas and partially selected elements based on the respective criteria. In contrast, (Equation 40)–(Equation 42) incorporate Algorithms 1–3 with low resolution DAC and DPC-enabled transmitters.  
**Algorithm 1:** Initial access-based optimal number selection algorithm.
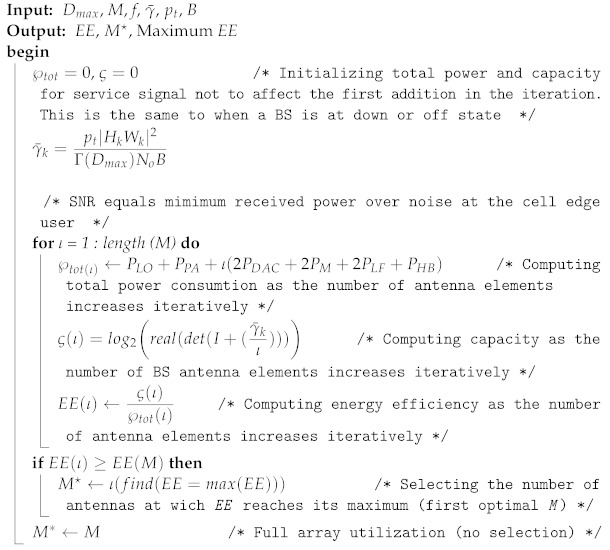

**Algorithm 2:** Number and element selection after reduced distance.
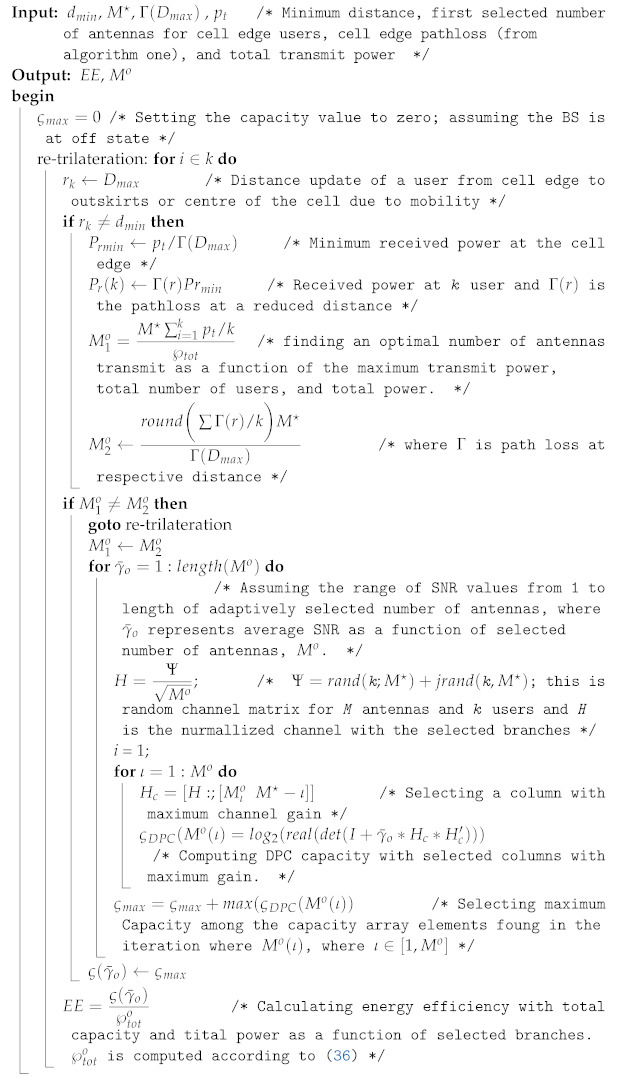

**Algorithm 3:** Proposed algorithm for low resolution DAC.
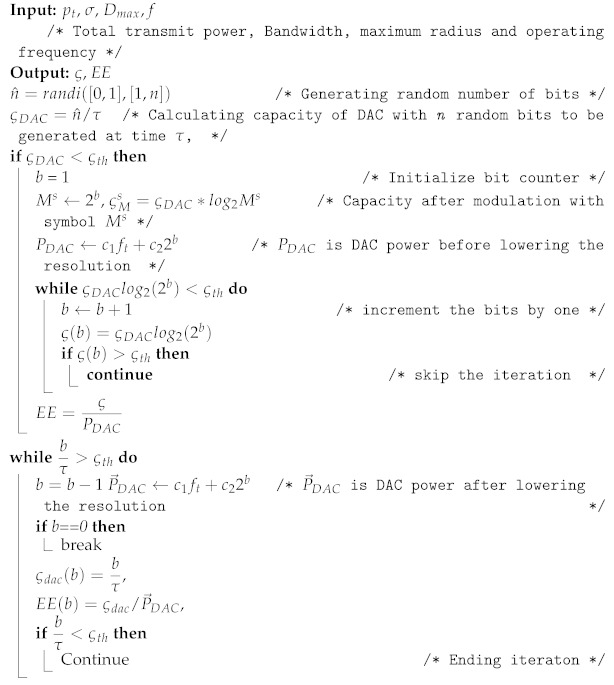


The procedures of the above two algorithms are summarized as follows:In algorithm one, the selection is made with dynamic or nondeterministic channel conditions. The input for selection is a pilot signal, and the number is identified based on the optimal value $M^{⋆}$ of the EE curve.In algorithm two, the maximum number of antennas $M^{⋆}$ is transformed to $M^{o}$, which was obtained as a new number in algorithm one.After identifying the minimum received signal at the cell edge and an optimal number of antennas ($M^{⋆}$) using the initial access condition in algorithm one, the number of antennas is adaptively reduced. Instead of reducing the transmit power when users move from the cell edge area to the cell centre positions, in this case, the number of antennas is reduced, maintaining the minimum required SNR for connection quality. Then, after finding the average SNR, the number is changed from $M^{*}$ to $M^{o}$.After identifying $M^{o}$, the branches with best channel gains are selected from $M^{⋆}$ iteratively, and the DPC capacity is computed.

The general idea behind algorithm three is stated as follows:Assuming fixed SNR at the cell edge, the minimum capacity is obtained and considered as a threshold capacity. Here, the free space propagation model is considered, and small-scale fading is ignored.Through randomly generated bits, the capacity of DAC is calculated and compared with the threshold.If the capacity of DAC, which was obtained with random bits, is greater than the threshold capacity, then DAC down resolution is done by iteratively decreasing the number of bits before analogue conversion is done. This reduction in bits ultimately reduces the operating power consumption of DAC according to (Equation 14). This DAC power directly affects the total system power consumption according to (Equation 37).Finally, the EE is computed as a function of the capacity and total power with low resolution. Throughout the evaluation process in this paper, the following parameters in Table 2 are used.

## 5. Results and Analysis

In this section, the simulation results with different scenarios are discussed. Figure 2 illustrates the relationship between distortion and the number of bits or symbols. As the number of bits or symbols increases, distortion decreases logarithmically, and this accounts for the increase in bit resolution. As shown from the figure, the distortion reaches its minimum point when the number of symbols is 7 when the evaluation is at the bit level, and it lasts until around 65 at the symbol level. Figure 3 shows the effect of the number of symbols on quantization. Quantization is the inverse of distortion, which increases logarithmically with the number of symbols. As the number of bits is mapped to each constellation point in any digital modulation scheme, the number of symbols increases exponentially and directly affects the quantization level.

The relation between DAC power consumption and energy efficiency is illustrated in Figure 4. The EE is evaluated through a range of power values from 0.3 to 3.48 mW. As the power consumption of the DAC increases due to an increase in the number of bits entering to DAC as in (Equation 14); then, the energy efficiency decreases exponentially according to the EE equation in algorithm two. This happens if the increase in bits affects the total power consumption more than the capacity. Moreover, an increase in the total power consumption with maintained minimum capacity leads to a decrease in the EE. Again, the graph shows that applying low resolution in DAC lowers power consumption and so that the EE increases.

The relationship between energy efficiency and the number of antennas considering the low-resolution DAC and without resolution case is shown in Figure 5. As the number of antennas increases, the energy efficiency also increases, and when low-resolution DAC is applied, the energy efficiency becomes higher than without resolution. The peak EE with low-resolution is 14.8 and 10.2 Mbits/J with and without low resolution, respectively. The EE begins to decline after the peak point because the increasing number of antennas on total power consumption exceeds the increase in capacity, which is achieved due to the increasing resolution.

The trade-off between spectral and energy efficiency for the given number of users with and without low resolution is shown in Figure 6. Both spectral and energy efficiency increase together for a fixed bandwidth up to the optimal energy efficiency point. For this, we evaluated the energy efficiency for a different number of user scenarios; for example, *k* = 5 or 10. For a smaller number of users, the diminishing rate is faster than that of a larger number. Hence, the EE graph starts declining at 55 and 59 bps/Hz of SE for five users with low-resolution DAC and without, respectively.

High energy efficiency is achieved with low-resolution DAC for the given spectral efficiency. The maximum energy efficiency point is 14 Mbits/J, which is when the spectral efficiency reaches 115 b/s/Hz with ten users and without resolution. It becomes 10.5 Mbits/J for the same number of users and spectral efficiency. Figure 7 depicts the relationship between energy efficiency, *k*, and *M* in a massive MIMO system with statistical and instantaneous SNR values. For cell edge users in LoS conditions, the outcome is evaluated using procedures of algorithm one. While the energy efficiency increases with the increase of *M* at first, it begins to decline at some point as *M* continues to grow, according to the simulation.

In this figure, statistical and instantaneous or fixed SNR are also compared for the same *k*. It has been demonstrated that fixed SNR outperforms for small *M* and under-performs for large *M*. EE has also been shown as the number of user terminals increases. However, due to random channel conditions, EE exhibits different optimal points. Moreover, the optimal threshold of EE for each configuration varies according to the number of users and SNR modalities.

Figure 8 and Figure 9 present the results according to the proposed algorithm by combining the three scenarios, and we compared the performance of each at CI and FSPL using mmWave and sub-6 GHz frequency ranges. The first scenario entails locating $M^{⋆}$ from the entire array at the indoor cell edge, locating $M^{o}$ according to (Equation 23), and finally evaluating capacity values as (MMo) using combinational permutation. At initial access, equal power allocation among all BS antenna elements and the point at which the EE graph starts diminishing is evaluated using a reference signal. Then, the number of antennas is used as a baseline for our further considerations.

Before the energy efficiency evaluation process, we performed an analysis of teh free space and CI path-loss models according to their formulations stated in (Equation 19) and (Equation 20). Accordingly, the FS model provides higher data rates due to obstruction freedom; however, CI is more realistic than FS in practical scenarios. Based on this intuition, we applied an antenna selection algorithm to both, and the results show that a minimal number of antennas are selected in free space compared to CI.

When CI path loss is applied to mmWave and sub-6 GHz frequency ranges and compared for fixed total system power, CI with sub-6 GHz is more energy-efficient than mmWave. Although the high-frequency signal carries larger data than the low-frequency signal, as frequency increases, the blockage due to different impairments also exhibits low wavelength, which negatively affects the received signal. Low received signal accounts for a low data rate at the receiver, so EE is degraded compared to CI. Finally, we found that the FS path loss with the DPC precoder changes the graph from a logarithmic to almost linear because only a few antenna elements were selected compared to CI.

The effect of the transmit power on the EE is depicted in Figure 9. We evaluated EE as a function of BS antennas at different power levels for full array and selection implementations. The system’s performance was also evaluated with and without the non-linear preceding, which showed that antenna selection with the minimum SNR significantly improved the energy efficiency with less transmit power and a DPC precoder.

Figure 10 illustrates EE with a number, complex, and random element selection and finally compares it with full array utilization or no selection. It can be observed that random number selection followed by complex selection shows better performance than no selection or full array. However, when it is compared with the number of selections made with our proposed algorithm and complex selection, random selection still outperforms for a smaller number of antennas employed and under-performs for large numbers of antennas.

The relationship of energy efficiency with low-resolution DAC and selection are shown in Figure 11. The EE can be enhanced by applying a low-resolution DAC algorithm even if the full array is utilized. We also evaluated the random selection after finding the optimal antennas and applying low DAC resolution. The results show that applying low DAC resolution still enhances EE as it has a significant role in minimizing the total power consumption.

Figure 12 presents the complex nature of the proposed selection algorithm and compares it with selected literature that used similar techniques. Complexity, in this case, is the number of iterations primary and nested loops to happen while selection is made to identify the branch with better channel gain among the entire array. Hence, the result also illustrates the system’s complexity when selection incorporates low-resolution DAC according to table one. From the graph, we can observe that random selection is the least complex even though it has a lower capacity than complex selection. This is because the selection is made irrespective of the channel gain, which plays a crucial role in enhancing the capacity and complexity. For random selection, the number of iterations to select *M* antennas is only one as it has no combination with the channel branches.

Our proposed algorithm is also compared with [21,28,29], which are among the simplest and following similar approaches to the best of our knowledge. The complexity order of each is [21] our proposed technique [28,29] and random selection according to (Equation 27) and (Equation 28). We also found that the proposed algorithm is more energy-efficient than random at the cost of some complexity, which is less than that of [21].

Moreover, since the energy efficiency of the proposed technique has been shown to surpass random selection and full array utilization or no selection in Figure 11. Figure 12 is to show only how complexity costs while working for energy efficiency; however, the rate of the effect and trade-off, including the EE of the aforementioned literature, is left as future work. Therefore, the selection technique meets our main goal of proposing an energy-efficient system at the cost of some complexity.

## 6. Conclusions

In this paper, energy-efficient antenna selection techniques were presented. In particular, we proposed adaptive transmit antenna selection strategies for downlink systems to minimize the power consumption of RF chain components associated with each antenna element. The selection process was categorized into two parts to reduce the complexity arising from several iterations. In the first part, we considered only cell edge users and found the average minimum SNR value from multiple generations of random channels to find the number of antennas at which the EE curve reached its maximum and began declining.

The optimal number of antennas obtained through this process was used as a baseline for further selection while users moved to the centre of the cell. The selection depended on the distance and channel condition between the users and a BS in the second case. The number of antennas to be selected adaptively changed with channel and user distance variations. We also proposed low-resolution DAC to further minimize the system’s total power consumption to enhance EE.

We evaluated the system’s performance at sub-6 GHz and mmWave frequencies with CI and free space propagation models. Furthermore, we compared the proposed antenna selection with and without low-resolution DAC. The results show that selecting only a few antennas instead of employing all the arrays improved the EE by reducing the total power consumption. Furthermore, we demonstrated that applying non-linear precoders, such as DPC, further improved the EE by enhancing the system’s capacity. However, the combined average EE was found to surpass selection without low resolution at the cost of some complexity.

## Figures and Tables

**Figure 1 sensors-22-01743-f001:**
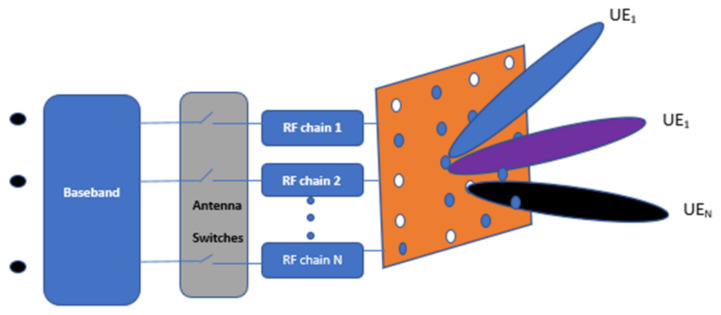
Digital beamforming URA configuration.

**Figure 2 sensors-22-01743-f002:**
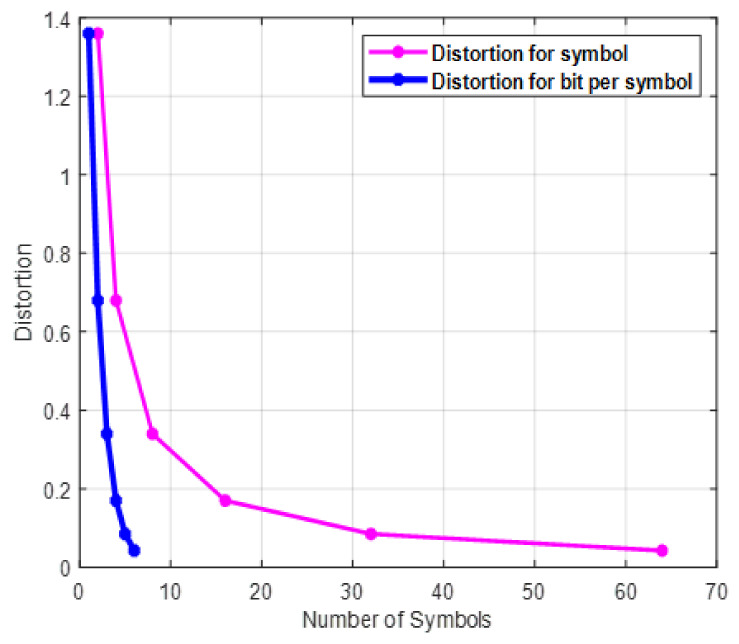
Distortion versus number of symbols with *k* = 5 or 10 and *M* = 64.

**Figure 3 sensors-22-01743-f003:**
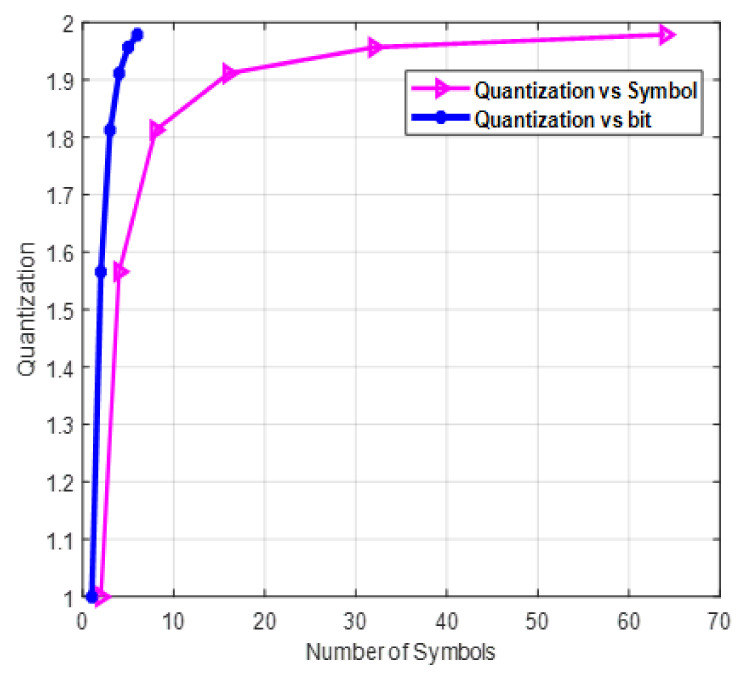
Quantization versus number of symbols with BPSK modulation.

**Figure 4 sensors-22-01743-f004:**
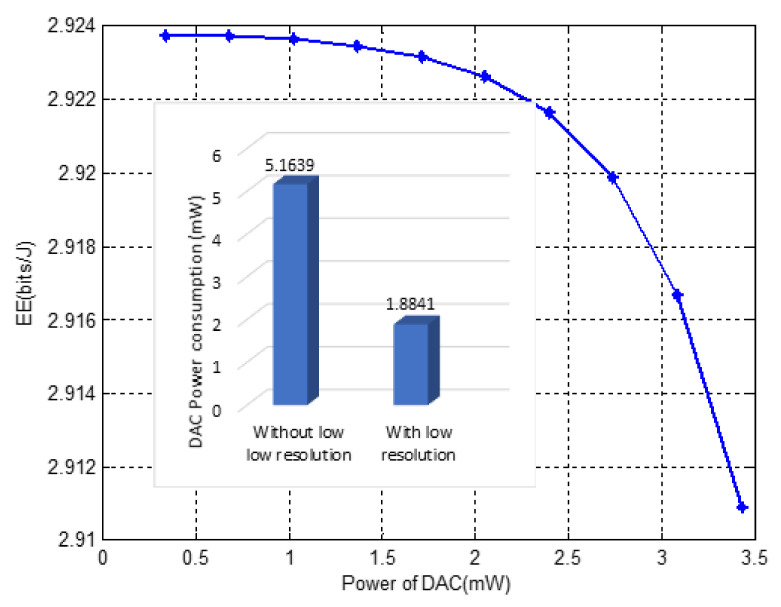
EE evaluation with different DAC power values, and evaluation of the total power consumption with and without low DAC resolution at randomly generated bits.

**Figure 5 sensors-22-01743-f005:**
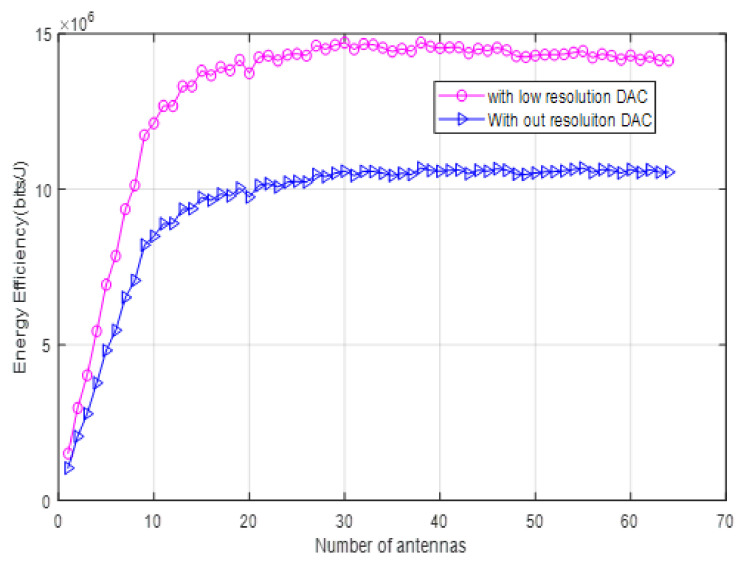
Energy efficiency as a function of the number of BS antennas when the number of terminals *k* = 30.

**Figure 6 sensors-22-01743-f006:**
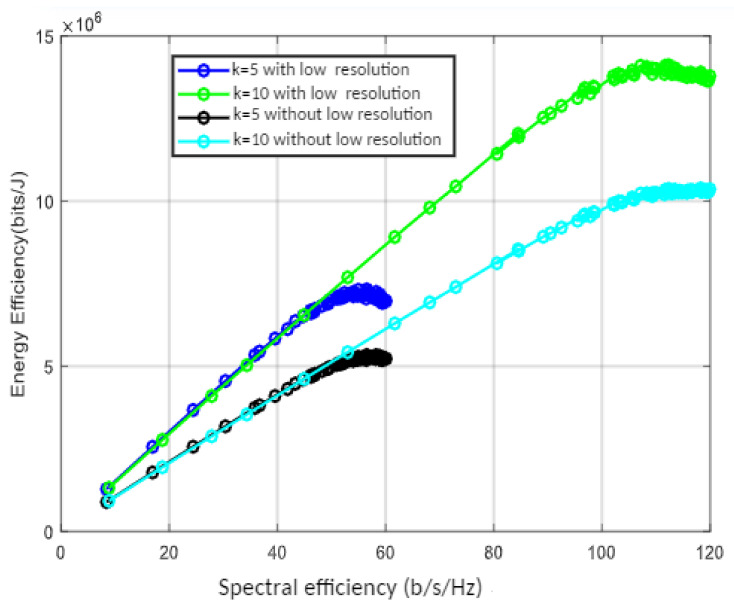
Energy efficiency versus spectral efficiency with without low resolution DAC.

**Figure 7 sensors-22-01743-f007:**
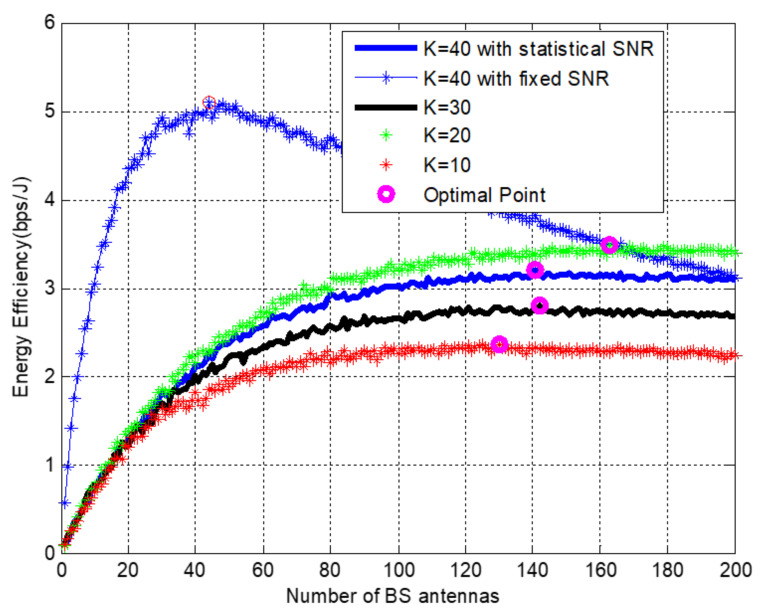
Energy efficiency as a function of *M* base station antennas with $P_{t}=20$ mW.

**Figure 8 sensors-22-01743-f008:**
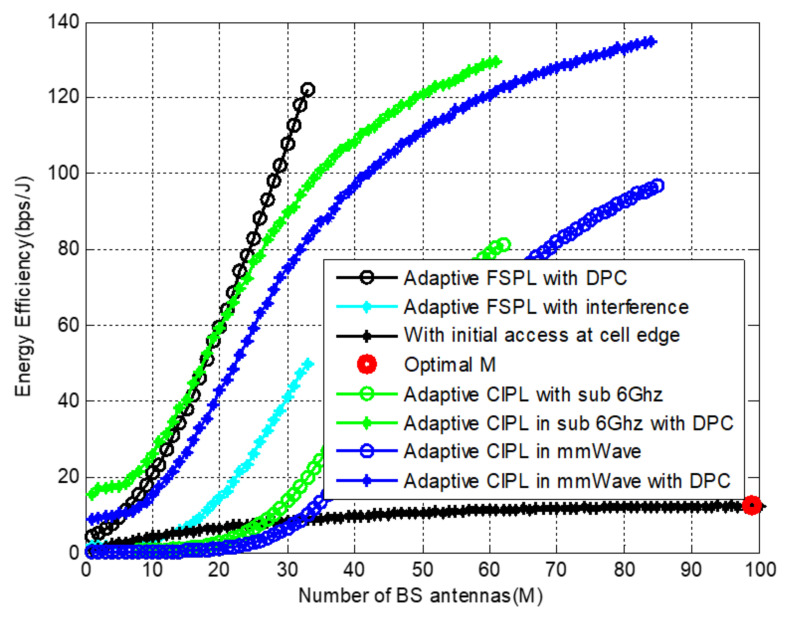
Energy efficiency evaluation as a function of the number of BS antennas with at mmWave frequency, *f* = 38 GHz and sub-6 GHz, and *f* = 2.5 GHz.

**Figure 9 sensors-22-01743-f009:**
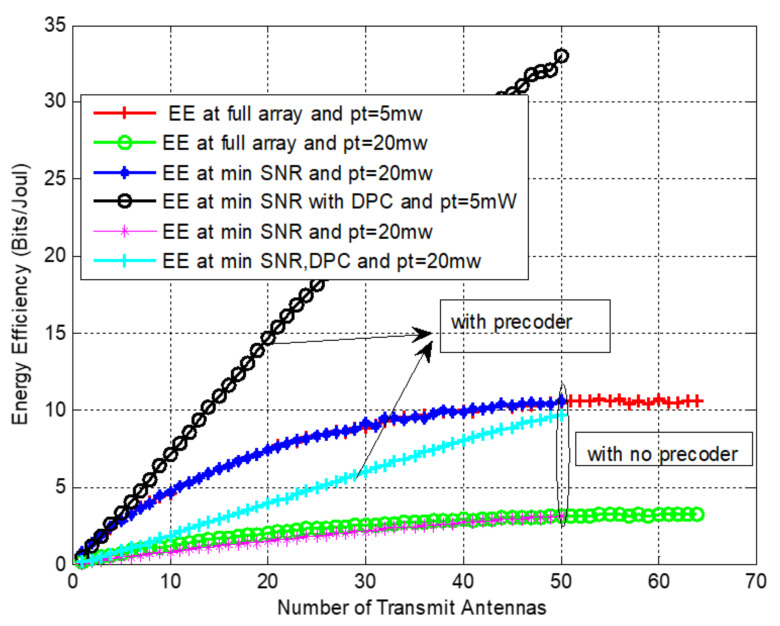
Energy efficiency evaluation as a function of number of BS antennas with at mmWave frequency, *f* = 38 GHz, and *M* = 64.

**Figure 10 sensors-22-01743-f010:**
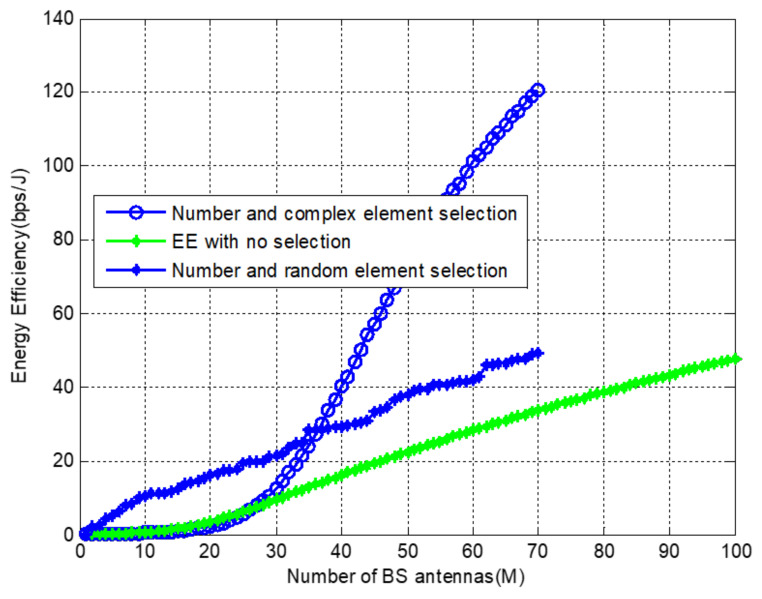
Energy efficiency evaluation with random and complex selection at *f* = 38 GHz and *M* = 100.

**Figure 11 sensors-22-01743-f011:**
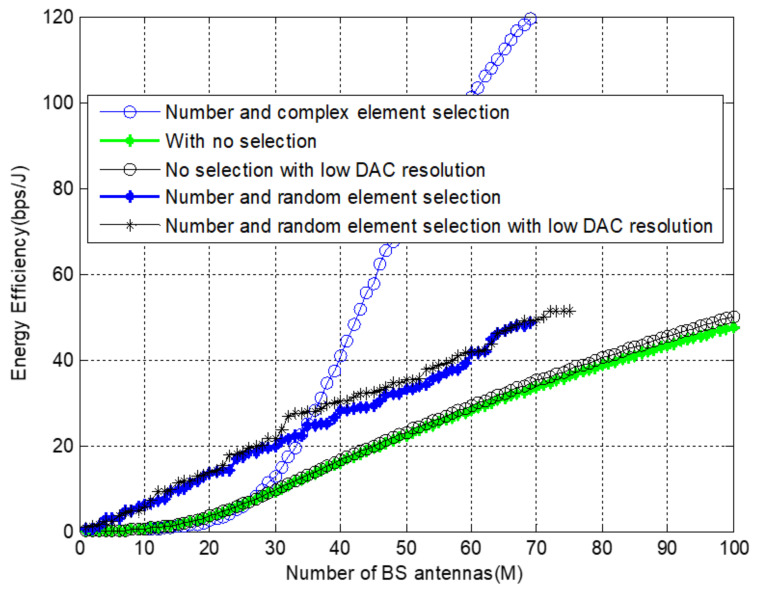
Energy efficiency with and without low resolution DAC at *f* = 38 GHz and *M* = 100.

**Figure 12 sensors-22-01743-f012:**
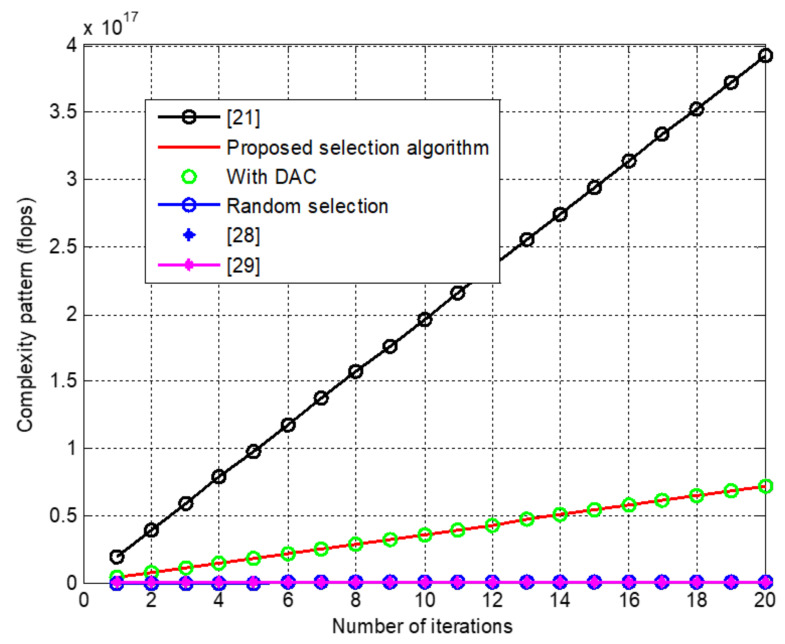
Computationalx complexity of selection algorithms with adaptive $\ell^{⋆}$ and *M* = 64.

**Table 1 sensors-22-01743-t001:** Complexity level measurement of the algorithms.

Algorithm 2	Algorithm 1 + 2	Algorithm 1 + 2 + 3
n^(MMo)	n^(MMo−ℓ⋆)	n^(MMo−ℓ⋆)+⊔

**Table 2 sensors-22-01743-t002:** Simulation parameters.

Parameter	Description	Value
$c_{1}$	Static power of DAC	$9\times 10^{-12}$
$c_{2}$	Dynamic power of DAC	$1.5\times 10^{-5}$
*B*	Channel bandwidth	40 MHz
$D_{max}$	Cell edge distance	200 m
$d_{min}$	Minimum distance	3 m
*f*	Operating frequency	38 GHz
$\bar{\gamma }$	Total average SNR at the cell edge	$\frac{Pr_{min}}{N_{o}B}$
*c*	Speed of light	$3\times 10^{8}$ m/s
$\Gamma (D_{max})$	Path loss at cell edge distance	${\left(\frac{4\pi D_{max}f}{c}\right)}^{2}$
τ	Random bit generation time interval	5 ms
$p_{A}$	Amplifier power	0.05 mW
$p_{M}$	Mixer power	0.04 mW
$P_{LO}$	Local oscillator power	0.01 mW
$\chi_{\hat{\sigma }}^{CI}$	Large-scale signal fluctuations due to the CI pathloss model	4.4 dB
$P_{LP}$	Low pass filter power	0.012 mW
$P_{HB}$	Hybrid with buffer power	0.033 mW

## Data Availability

Not applicable.

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
