# Peer review of "Hardware Efficient Massive MIMO Systems with Optimal Antenna Selection"

_sensors, 2022, doi:10.3390/s22051743_

Round 1

Reviewer 1 Report

please find the attachment

Author Response

Dear Sir/Madam,

I hope this mail finds you well. We have revised the paper with the appropriate corrections as per your comments. Besides, we have improved some grammatical errors by proofreading. Finally, we have attached the responses with your comments for detail, and we look forward to hearing from you soon.

Best Regards,
Shenko

Reviewer 2 Report

Please check the attached pdf review file.

Author Response

I hope this mail finds you well. We have revised the paper with the appropriate corrections as per your comments. Besides, we have improved some grammatical errors by proofreading. Finally, we have attached the responses with your comments for detail, and we look forward to hearing from you soon.

Best Regards,
Shenko

Round 2

Reviewer 1 Report

In this revised manuscript, the authors have addressed some concerns of mine. However, there are still many problems that the authors have not solved reasonably. Here are the details of the remaining concerns:

  1. Then, in (2), (3), (4), (5), there is the index k. What does k mean in the equation that defines the channel between w-th user and m-antenna? The authors claimed that the index w is used to represent the index of the considered antenna element of user k. But in this manuscript, there is only one antenna at each user, so w is always equal to 1.You can not use a constant to represent a changing index. Furthermore, if you want to keep the notations of the general case, you should use the notation like h_{k,1}. In summary, the authors must select between k and w to represent the index of user and use this notation consistently. w and k cannot be used interchangeable like in the current manuscript.
  2.  In (4), ZoD, AoD are written as the variables of the function \tilde{A}, but in the following paragraph, they are just the subscript of \beta and \delta. Which one is correct?The authors said that only k is in the subscript. But then which quantities do ZoD, AoD represent? I think they should be in the subscript too.
  3.  What does N relate to M? Why don't use a consistent notation? The authors explained that N represents the number of RF chains, while M represents the number of antennas. But the explanation does not appear until at page 6. In the previous pages, M and N have been used arbitrarily, which makes a big confusion in the mathematical model. Furthermore, what is the benefit to use 2 notations that are equal to each other? In the whole paper, the authors assume that M = N, so I strongly recommend that only one is used consistently in the entire manuscript. This would satisfy the standard representation of mathematics.
  4.  Why does the data symbol of k-th user have k entries? The authors' explanation does not satisfy the Reviewer. Again, if the authors only consider a special case, they should use the notations corresponding to this case, but not the general notations. Furthermore, if \dddot{d} has k entries, it cannot be multiplied with b_k, which has the size of N x 1.
  5.  In (13), H_{N} is undefined --> the authors has describe it in the revised manuscript but the size of it is still described yet. And what is the difference of H_N and H_k? If the number of user is greater than N, then there must be the case k = N, which causes confusion.
  6.  Why is the interference term in (16) ignored when computing the capacity? --> the authors have explained this to the Reviewer but have not added this explanation in the revised manuscript.
  7. Line 232: \iota \in [1,M]. Does it mean \iota can be rational? --> the authors' response is not what I concern. In mathematics, \iota \in [1,M] means that \iota is a real number in the interval [1,M], i.e. 1 ≤ \iota ≤ M. But in the manuscript, as my understanding \iota is an integer and cannot be any number in [1,M]. For example, \iota cannot be 1.23. The authors should change it to \iota \in {1,2,...,M}.
  8.  In the line after Table 1, what does M^{o2} means? Does it mean {M^{o}}^{2}? (the authors have addressed my concern but it causes another concern).
  9.  What does H mean in (29)? --> the authors have defined it, but now it is confused between H, H_N, and H_k. Please clarify. 

The reviewer also appreciate the authors' effort in revising the manuscript and found out other confusion and errors that the Reviewer has not found.

Author Response

Dears,

I hope this mail finds you well. First of all, I would like to thank the reviewers for their lovely comments and suggestions, which we have learnt a lot from. Accordingly, we have updated the manuscript based on the comments, and we look forward to hearing good news from you. 

Best Regards,

Shenko,

corresponding author

Reviewer 2 Report

The revision has been made on the basis of the reviewer comments. 

Author Response

Dears,

I hope this mail finds you well. First of all, I would like to thank the reviewers for their lovely comments and suggestions, which we have learnt a lot from. Accordingly, we have updated the manuscript based on the comments, and we look forward to hearing good news from you. 

Round 3

Reviewer 1 Report

The authors have corrected the mistakes and errors in the mathematical presentation that I have mentioned in the previous review. I have no further comment, except a suggestion that the authors should proofread their manuscript carefully to find the possible remaining errors before submitting the final version.

This manuscript is a resubmission of an earlier submission. The following is a list of the peer review reports and author responses from that submission.

Round 1

Reviewer 1 Report

This paper evaluates low resolution DAC and transmit antenna selection to reduce total power consumption and achieve energy efficiency in massive MIMO with less complexity.

1) Explaining many abbreviations used in the paper is insufficient. Abbreviations should be placed where they first apprear and used consistently. ex.) DAC (page 1) EE, BS (page 2) ...... etc.

2) The authors said that there are few prior studies for antenna selection in mMIMO systems. We could find many references in the literature for antenna selection in mMIMO systems. It is suggested to find and include them.

3) It is not clear to follow all symbol notations throughout the paper. The use of all symbol notations is very poor. It is one of the biggest hazards to understand the paper. The authors should improve symbol notations and their definition and description. Please rewrite them.

4) It is not easy to follow and understand every steps in Algorithms 1, 2, and 3. It is also one of the biggest hazards to understand the paper. It is suggested to include more clear explanations and improve the descriptions. ex) symbol p_tot(l) related part in Algorithm 1

5) In the simulations, the reader cannot identify the values employed for all channel parameters in the simulations.  

6) In Figure 12, complexity comparison results need to further be explained. It is unclear what complexity expresssions are used for comparison.

Based on the above comments, the paper should require major revision.

Author Response

Dear reviewer, I have attached the responses of authors for each comments. Please find herein attached word document file.

Kind Regards,

Shenko

Reviewer 2 Report

In this manuscript,  the authors present a low resolution Digital to Analogue Conversion (DAC) and transmit antenna selection at the downlink of a massive MIMO cellular wireless system to reduce the total power consumption and achieve better energy efficiency. The authors use the Close In (CI) path loss model for mmWave bands for their evaluation.

The proposed idea sounds good but I cannot verify the correctness of the proposed algorithms or evaluate the contribution of this work because of the poor mathematical presentation in the manuscript. In particular, many mathematical symbols are provided without any description. Some mathematical notations cause confusion.

This manuscript is written in good English grammar, but there are some sentences that are vague or confusing in meanings.  

 The followings are my comments in details:

  1. The abstract is quite long and does not focus on the main point. In fact, the first 4 sentences of the Abstract should be moved to the Introduction section.
  2. In the abstract, the authors state that they evaluate the downlink but in the body of manuscript, they mention about uplink transmission several times. So, I don't understand which direction they are considering.
  3. I don't understand what the authors mean when saying " ... reducing total power consumption and achieve energy efficiency in mMIMO at the cost of less complexity". Is "less complexity" a cost?
  4. I also do not understand this sentence " Although our selection algorithm also shows that when partially applying the same exhaustive search technique, the complexity and energy efficiency decrease and increase respectively due to double selection before exhaustive search".
  5. In the system model description, the authors state that there are M antennas and K users, but in Figure 1, there are N RF chains. Please explain.
  6. Mathematical symbols are not provided in a scientific standard. In particular:- Many symbols come without description: Pk (there is description for \( \hat{P}_k\) but not Pk), \hat{k}, Φ, z, Nt, FRF, D, and many more.- Many symbols are not written in correct form. For example, the number of user K and the kth user are usually presented by the same lowercase letter k, the matrix c should be presented by capitalized letter (C), κ and k are used confusingly, etc.- I think (6) is not correct. It should be [hk1, hk2, ..., hkm].- There are too many symbols. Some symbols are very simple, for example, ζ = 2π/λ, why do the authors need to introduced the new symbol ζ?- The operator ∑ appears without the index and limits. - "For i ∈ k, ..." (algorithm 2) --> is k a set or a number?
  7. There is not any proof for the mathematical results. I wonder if they are trivial or too complicated to present.
  8. I have a suspect on the feasibility of the proposed algorithms because I cannot know which constraints have been introduce and whether they are sufficient.
  9. Many quantities are introduced in the mathematical model, but I don't see the introduction about the simulation setup, i.e. how the simulation parameters are selected and what their values are.
  10.  Why the values of DAC power in Figure 4 are presented in a downward order? 

Author Response

Dear reviewer,  

Greetings! I am sending the  responses for each comments. Please find herein attached word file.

Kind Regards,

Shenko

Round 2

Reviewer 1 Report

The review pdf file is attached.

Reviewer 2 Report

In the previous reviewing, the most important concern of mine is that the presentation (both mathematical presentation and wording) of the manuscript is not clear enough for the readers to understand what the authors are doing.

In the revised version, the authors have made some effort to address my concerns but the main concern mentioned above is still not addressed sufficiently. In particular, what the authors did is mainly replying to my questions, but in fact, there is lacking of significant changes in the manuscript. Therefore, I still cannot approve this version of the manuscript.

In particular, I will discuss point by point of what I have commented in the previous review and whether the response of the authors is sufficient:

1. The abstract is quite long --> it has been improved by the authors.

2. Do the authors consider the signal transfer in uplink or downlink direction? --> The explanation is not clear and there is no action in the revised manuscript.

Even if the channels are reciprocal (actually I don't see any assumption about the reciprocity of the channels in the manuscript), there would be some difference between considering the signal models for downlink or uplink transmission.  In particular, for uplink, the SINR is computed at the mobile terminal user and it depends on the transmit power of the BS, while for downlink, the SINR should be computed at the BS and should depend on the transmit power of each mobile user.

In summary, please take action about the "uplink - downlink confusion" in the revised manuscript.

3. About the confusion of "less complexity" --> the authors have corrected it.

4. About the confusion on "exhaustive search" --> the authors have improved the description of "exhaustive search" but it is insufficient. There is still confusion in the rewritten paragraph. In particular:

-  "some antennas are deducted from the entire array using reference signals" --> which reference signals? and how can the antennas be deducted? It's should be better to refer to a specific algorithm in the analysis section.

- what does "user distance-based selection" mean? Please clarify. 

- "our selection algorithm incorporates the exhaustive searching method
to select an element with the best channel gain
"  --> do the authors select an element or multiple elements? In the response, the authors give an example of selecting 4 best antennas, so I think this should be multiple-element selection.

- In the previous paragraph (lines 79 - 92), the authors discussed a three-part procedure. Is there any related between this three-part procedure with the two-phase selection mentioned above?

- Also in the paragraph from lines 79 to 92, the notation M0 has not defined yet. What is it?

5. Confusion in the symbol notation (N or M)? --> The authors have replied my questions but there is no action in the revised manuscript. Please use consistent symbol in the manuscript and in the figures.

6.   Mathematical symbols are not provided in a scientific standard. --> This is still a big problem in the manuscript. There are many symbols that are not defined, such as M0, N or M in the previous comment. Many subscripts are not put in their correct position, such as bi, δk,AOA, etc.

I consider this the serious weakness of the manuscript. Please spend time for a thorough proofread of the manuscript.

7. There is lacking of mathematical proof. --> Again, there is a short reply from the authors but there is no action to revise the manuscript.

8.  I have a suspect on the feasibility of the proposed algorithms because I cannot know which constraints have been introduced and whether they are sufficient. --> I agree with the reply from the authors but again I didn't see where in the manuscript that the constraint on capacity is expressed mathematically.

9. Simulation setup is not described. --> The authors have addressed this concern quite well. But I still have some other comments about this issue:

- In Table 2, the writing of the simulation values and its units are not consistent. For example, "40 MHz" and "38GHz" (space or no space between?). Furthermore, the multiplication symbols are confused with letter "x".

- Many input parameters are not introduced in the description of Algorithms 1, 2, and 3. This makes the Algorithms hardly readable by the readers.

10. The x-axis values in Figure 4 are arranged in downward order. --> I don't think the reason provided by the authors is convinced (at least by me). It's very strange to a mathematician to see a plot in which the horizontal axis is reversed. If you prefer to do that, you should use other tools such as the bar chart.